# Views of Moroccan University Teachers on Plant Taxonomy and Its Teaching and Learning Challenges

Lhoussaine Maskour [1,2,*], Bouchta El Batri [3,4], Jamal Ksiksou [5], Eila Jeronen [6], Boujemaa Agorram [7], Anouar Alami [4] and Rahma Bouali [8]

1   Regional Center for Education and Training Professions (CRMEF), Oued Eddahab, Dakhla 73000, Morocco
2   Laboratory of Science and Technology Research (LRST), ESEF, Ibn Zohr University, Agadir 80000, Morocco
3   Regional Center for Education and Training Professions (CRMEF), Fes-Meknes, Fez 30000, Morocco
4   Engineering Laboratory of Organometallic, Molecular Materials and Environment, Faculty of Sciences, Dhar El Mahraz, Sidi Mohamed Ben Abdellah University, Fez 30000, Morocco
5   Sociology-Psychology Laboratory, Faculty of Letters and Human Sciences, Dhar El Mahraz, Sidi Mohamed Ben Abdellah University, Fez 30000, Morocco
6   Faculty of Education, University of Oulu, 90014 Oulu, Finland
7   Interdisciplinary Research Laboratory in Didactics, Education and Training (LIRDEF), Ecole Normale Supérieure, Cadi Ayyad University, Marrakech 40000, Morocco
8   Laboratory of Informatics Signals Automation and Cognitivism (LISAC), Faculty of Sciences, Dhar El Mahraz, Sidi Mohamed Ben Abdellah University, Fez 30000, Morocco
*   Correspondence: lhomaskour@gmail.com; Tel.: +212-670-283-216

**Abstract:** Plant taxonomy includes the identification, description, and classification of plants at the level of species or other taxa. This study aims to analyze the views of university teachers on plant taxonomy and its teaching, the causes of the shortage of plant taxonomists, and the challenges encountered by students in learning plant taxonomy. University teachers in Morocco ($n = 24$) responded to a survey consisting of fixed and open-ended questions. The data was analyzed by inductive and deductive content analysis. The results showed that all university teachers considered a taxonomist as a scientist and plant taxonomy as a dynamic and highly scientific, and descriptive discipline. They stated that the taxonomist community is in crisis because of the shortage of plant taxonomists and the decrease in the quality of training provided at the university. The biggest challenges in learning plant taxonomy were the prevalence of traditional teacher-centered methods, the inadequacy of time and didactic resources spent on teaching, and the Latin nomenclature. The difficulties associated with the concept of evolution and the diversity of classifications were also mentioned. The angiosperm group was the most difficult for students to understand. Furthermore, this research shows that the financial, human, institutional, pedagogical, and didactic resources for the teaching of plant taxonomy are insufficient and do not allow for the use of teaching methods supporting learning. How to plant taxonomy is taught is important, and when it is considered difficult, it can lead to a reluctance to study plant species and be one of the reasons for the decline in plant taxonomists in Morocco. Consequently, this issue can negatively affect the preservation and conservation of local flora.

**Keywords:** university teachers; inductive and deductive content analysis; teaching and learning plant taxonomy

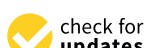



## 1. Introduction

Plants represent the basis of most terrestrial ecosystems. To protect endangered species, it is essential to establish a complete inventory of plant life before their extinction [1]. Plant taxonomy includes the activities to identify or classify organisms; it is a dynamic science adapted according to the data collection of taxonomists [2,3]; its objective is to inventory all existing forms of life and describe their specific characteristics [4]. Botany has societal

concerns, such as biodiversity preservation, food security, sustainability, and climate change; its methods need to be recognized [5–7].

Due to the rapid loss of plant life and its consequences on humanity, botanical education deserves a greater role in education [8–10]. Plant taxonomy is a necessity for naming different plants and establishing rules for their identification. This key information is used for determining the relationships and place of a plant in an ecosystem and its status in preservation and conservation programs [2,11]. Identifying and classifying plants is a challenge for taxonomy experts, but it is also a challenge for students.

Over the past two decades, students' interest in plants and plant identification has declined significantly, leading to a reduction in their knowledge about plants, plant characteristics, and plant identification and use [12–15].

At the same time, a good deal of information on plants that is important in terms of biodiversity and sustainable development has been lost [14,16,17]. Similarly, the teaching of plant taxonomy and its university research, as well as the number of its experts, have decreased in recent years [9,18,19]. This decline is affecting the quality of botanical practice and the training and establishment of the next generation of botanists [5,6].

In Morocco, there is a disaffection of students for the specialties of botany and plant taxonomy, which poses the problem of renewing the potential of researchers and experts in this field [20]. However, it is very important to have knowledge of and interest in species, species identification, and plant taxonomy for several reasons. For example, without knowledge of species, it is impossible to understand the structure and function of ecosystems (life-supporting systems on the Earth) and the principles of biodiversity and its role in sustainability [15,21,22]. The aim of this study is, therefore, to elucidate the views of Moroccan university teachers on plant taxonomy and the difficulties they have encountered in teaching and the challenges to students' learning.

## 2. Literature Review

### 2.1. Plant Taxonomy as a Biological Science

Taxonomy is a discipline of biological science discipline with the aims of naming, identifying, describing, and classifying organisms at the level of species or other taxa. It is a dynamic science that is developed based on the data collection of taxonomists [23]. In plant taxonomy, plants are identified and classified based on their nature and taxonomic characteristics to determine their kinship [2]. Beginners usually define plants based on their morphological characteristics. In order to identify and name a plant or a plant group, an individual must be familiar with its morphological characteristics, and botanical terminology as well as have identification skills [2,24,25].

Today, taxonomists use molecular taxonomy in addition to morphological methods [26]. Indeed, the characteristic molecular structure of the oils and aromatics of plants had a very significant interest in their classification, or grouping them in different taxa in a more rapid and precise way. Thus, taxonomists should develop an understanding of molecular biology because if understanding molecular biology becomes widespread, the techniques of taxonomy will be used more frequently, and the identification of species will be more efficient than conventional techniques, and consequently, the programs for maintaining biodiversity and preserving the environment will be successful. [27]. The biodiversity crisis demands practical solutions to problems, many of which derive from taxonomy [28]. Thus, the role of taxonomy is to reveal biodiversity and promote the optimal protection and utilization of existing biodiversity for the benefit of current and future generations [27], and it is fundamentally important in ensuring the quality of life of future human generations on the earth [29]. This goal can be achieved through an education that introduces students to plant taxonomy and fosters a sense of the importance of biodiversity and their own responsibility to promote it. Botany is a thousand-year-old science—it deals with many societal issues, such as the safeguarding of resources, sustainable development, climate change, and the conservation of biodiversity. Botanical education is also becoming a priority in any education system [4,9,10]. The main objective of plant taxonomy is to

name and identify different plants [30]; it is a necessary step to know their use and the modes of their conservation. Hence, the identification and classification of plants remain a challenge for expert taxonomists and for teachers as well as students [11,26].

In Morocco, as elsewhere in the world, the situation of plant taxonomy is alarming: few specialists, with little means and motivation to flourish and, moreover, little or no valorization of their skills. It is extremely difficult to give a precise list of taxonomists, but it seems clear that human expertise in this regard remains insufficient [31]. The decrease in the number of taxonomists is a major factor that limits the knowledge of Moroccan biodiversity for the process of its conservation and sustainable use [31]. Moreover, addressing the exploitation, valorization and protection of national plant heritage still require a good deal of knowledge [32].

However, taxonomy is really not a national priority in scientific research [31], Morocco's floristic research is declining, and the worst is there is no training of young systematic botanists in the country [7,33]; in addition, the first-generation national researchers is ready for retirement with no relief, at least in the short term [7]. The practice of taxonomy in Morocco is affected by four main constraints: the lack of human resources, the lack of material and financial means, the lack of awareness of the importance of taxonomy on economic and social plans, and the lack of synergy between the actors concerned [31]. The lack of means and equipment concerns almost everything, from field equipment to drawing material of dissected parts, including everything necessary for sampling, conservation, storage [31]. According to Heywood [34], Morocco has four main botanic gardens; their objectives include environmental education and the conservation of native and endangered plants of Morocco. They are the botanic garden of the "Institut Scientifique", the botanic garden of the Hassan II Agronomy and Veterinary Institute, the "Jardin d'Essais Botaniques", and the exotic gardens of Bouknadel.

### 2.2. Teaching and Learning Plant Taxonomy

The skills to identify and classify plants are considered to be essential skills [27]. Plant taxonomy is taught at various levels, from basic education to higher education. In basic elementary education, the similarities and differences between plants and the grouping of plants are taught in the lower grades. In the upper elementary grades, students become more familiar with different plant species by using the external and internal structures of plants as a basis for grouping and classification. In upper-secondary education, different classification systems are taught, and in higher education, more detailed classification is based on the morphological, histological, or genetic characteristics of plants (cf. [15,22,35,36]).

Learning biological knowledge and thematic thinking requires an understanding of concepts, processes, phenomena and (hierarchical) structures, such as different levels of the cell, the genes or taxonomy. For many years, research has focused on the conceptions of higher education teachers about teaching and, to a lesser extent, about learning [37–40]. Like other teachers [41,42], university teachers have different views on teaching and learning [37]. In addition, their pedagogical choices vary depending on how they see teaching and learning. In addition, their views on teaching also differ depending on their experiences and ideas.

Several studies have shown that teachers [42,43] like to stick to traditional teaching. However, plant taxonomy has evolved rapidly with the development of science and technology [29,36]. Thus, in addition to the traditional teaching methods, such as teacher presentations, lectures and traditional group work [44], fostering new and innovative ideas is needed.

Studies show a positive relationship between outdoor education and students' knowledge of nature and positive attitudes toward nature [45,46]. In addition, repeated exposure to plants in a natural environment has been shown to be important for concept learning, and it is effective if the student complements the classroom exposure with effective study outside the classroom. Classroom time is not sufficient to learn reliable identification [30,47].

According to university teachers, the best and most effective method for developing identification skills is to study in an authentic natural environment [24]. The most

sustainable learning experiences in natural environments are achieved through student experience-based strategies [48]. Experiential learning is personal, sensory learning through a specific experience [49]. Fieldwork in the natural environment develops students' understanding of taxonomies [50–53]. It also develops students' cognitive learning [25] and observational skills [54]. The importance of experiences in the natural environment has been highlighted in several studies around the world [55–57]. However, teacher education programs rarely include practical methods for outdoor fieldwork [58].

In addition, computer and Internet-based learning offer opportunities for non-interactive to interactive student-centered learning experiences [29,30]. When students use simulations and modeling tools by studying phenomena outside the classroom [59], such as species identification and biodiversity, or when they use interactive tools, such as interactive concept maps, data representations, and timelines, they have the opportunity to create visual connections between their current knowledge and new ideas [60]. With the help of digital content production tools, they can create social and emotional connections with teachers, peers, the community and the rest of the world [59]. Cooperation learning can also be enhanced with social media content such as blogs, podcasts or with virtual presentations [61].

### 2.3. Contents Related to Plant Taxonomy in the Science Curricula at the Moroccan Universities (Life Sciences Section)

Among the 24 modules that make up the Life Sciences subject in Moroccan university curricula and during the three years of senior high school, some are related to plant classification. These modules are taught in lectures, tutorials and practical lessons. The students study at least three modules about plants, including at least one devoted to systematics. Moreover, the distribution and content of these modules during the three years of bachelor studies shows that students have a cell biology module and plant biology module in the first year, a plant biology and physiology module in the second year and in a third year. The fifth semester is the common core for the BCSs of all specialties in which students learn the biology of organisms and ecosystems (plant kingdom) and the theoretical and practical basics of botanical systems. The biosystematics module includes the history and principles of the classification in the plant kingdom and the classification and evolution of large plant groups, and the plant ecology module [62].

### 2.4. Challenges in Learning Plant Taxonomy

Many studies [62–67] show that despite the efforts made by teachers to improve the quality of the teaching and learning of plant taxonomy, this discipline constitutes a challenge for students. Thus, many misconceptions have been identified among students, which creates obstacles to learning plant taxonomy [62,64,68–72]. The studies carried out in France [69] and in Morocco [73] show that according to the secondary school teachers, in the official instructions as well as in several school manuals, the examples of plants cited in the educational activities relate to vascular plants (Angiosperms and Gymnosperms). This can generate erroneous conceptions among students, such as that biodiversity loss concerns only vascular plants and that plants of other taxa are not important [69,73]. University students, as well as secondary school teachers, also have significant difficulties in understanding the core concepts of botany that are prerequisites for the study of taxonomy [30,66,69]. Some students become confused regarding the definition of plant classification and its purposes, and consequently, they are unaware of the relationship between plant classification and the understanding of biodiversity and its preservation [53,71]. Students also have difficulties classifying various plant species, and thus, they confound fungi with plants and link some vascular plants and nonvascular plants, gymnosperm plants, to angiosperm ones. Students tend to classify plants according to recognizable characteristics (green color, growing in the soil) and different parts of the plant (stem, leaves, flowers, etc.) [9,22,35,62,70,72,74].

The persistence of these alternative conceptions is because the pedagogical approaches used in secondary education do not allow the development of the skills necessary to classify plants [30,36,62,66]; this should challenge us on the effectiveness of the current teaching of plant classification, which must be renewed [75,76].

### 3. The Aim of the Study and the Research Questions

The aim of this study is to clarify the views of Moroccan university teachers on plant taxonomy and its teaching, the issues concerning the shortage of plant taxonomists, and the challenges of learning plant taxonomy. The research questions are:

-Which kind of views do the university teachers express about plant taxonomy and its teaching?

-What are the causes of the shortage of plant taxonomists, according to university teachers?

-What kind of challenges do the students have in learning plant taxonomy, according to the university teachers?

The Moroccan results can be utilized for education concerning plant taxonomy teaching and learning worldwide because knowledge of species and species identification is not important only for plant taxonomists, but it is equally important for other people's nature experiences and their environmental attitudes and views on biodiversity and sustainability, all around the world (cf. [21,22,53,77–79]).

### 4. Research Methods

A mixed-methods approach using a questionnaire survey among university teachers was used.

#### 4.1. Sample

The participants were 24 university teachers; 5 women (20.83%) and 19 men. The respondents were invited to take part in the study on a voluntary basis. All of them were involved in the university's teaching of plant taxonomy, floristics, ecology, and plant biology. It is, therefore, a convenience sample; 87.5% of them have more than 20 years of experience teaching. Their specialties are biology and plant ecology. These specialties require strong knowledge and know-how in taxonomy techniques.

The participants are teacher-researchers from the following Moroccan higher education institutions: Faculty of Sciences Ben M'sik—Casablanca, Faculty of Sciences Agadir, Faculty of Sciences Dhar El Mahraz, Fez, Faculty of Sciences—Raat, Faculty of Sciences—Tétouan, Faculty of Sciences Semlalia—Marrakech, Faculty of Sciences Mohammed V—Rabat, Regional Center for Education and Training Professions—RABAT, Ecole Normale Supérieure—Marrakech, Agronomic, and Veterinary Institute Hassan II—Rabat, Scientific Institute—Rabat.

#### 4.2. Data Collection Tools

We opted to use a questionnaire as a data collection tool. To validate the collection tool, a pre-test of the questionnaire was carried out among seven university teachers. Four specialists in taxonomy and the didactics of biology analyzed the coherence and relevance of the questionnaire. After validation of the questionnaire, it was sent by email to 37 university teachers, of whom 24 responded. The questionnaire includes 25 questions dealing with the following four items: (Item 1: Conceptions and scientific nature of plant taxonomy; Item 2: Shortage of taxonomists/systematists; Item 3: Knowledge of institutional aspects; Item 4: Nature of learning difficulties in plant taxonomy and specific difficulties of each taxonomic group).

The questionnaire includes closed and open-ended questions. Respondents were invited to justify their answers whenever necessary. The questionnaire was administered face-to-face or by email after having received consent from each teacher. After the responses were collected, the data was analyzed using a combination of quantitative and qualitative methods.

The quantitative data was coded and analyzed using SPSS Version 21. The qualitative data were analyzed using inductive and deductive content analysis methods [80–82].

By organizing the data, the content analysis aimed to clearly present significant findings, identify various aspects and characteristics of content and present the significant findings [83]. Data analysis was carried out by two researchers using an inductive or deductive analysis method [80,84–86]. Both the answers of the teachers and the justifications of their answers were analyzed and categorized based on previous studies [67,69] by using deductive content analysis (e.g., question 8 relating to the scientific nature of plant taxonomy). In addition, the inductive content analysis method was used based on the categories created by the researchers themselves (e.g., the factors leading to the crisis of the taxonomic community), and in some analyses, both the inductive and deductive approach (e.g., students' challenges encountered on plant taxonomy teaching were subjected to inductive content analysis and students' knowledge levels, thinking skills and evaluation methods were subject to deductive content analysis) [80,87].

Researcher triangulation was an essential part of our analysis process [81]. To ensure reliability, the second researcher coded and analyzed data after the coding and the analyses done by the first researcher, then the findings were compared. When there were differences between the two researchers, a consensus was sought. If consensus was not found, the third researcher compared the views and suggested a solution which was negotiated by the researchers [82]. Because of the dialogical nature of the analyses, we did not see a need for calculating inter-rater reliability. Our research group consisted of experts in biology education and educational sciences and experienced teacher educators and researchers.

## 5. Results

### 5.1. University Teachers' Views on Plant Taxonomy

To find out university teachers' views on plant taxonomy, inductive and deductive content analysis was used [81,82]. To question 8 in the questionnaire ("Do you consider a taxonomist as a scientist?"), all respondents considered that a taxonomist is a scientist, but concerning the question "why", their responses were different.

The justifications of the university teachers were based on various reasons (Table 1): We noticed that the most mentioned justifications (29.5%) concerned the need for scientific tools and knowledge in taxonomy, and 23.5% focused on the use of knowledge from other scientific specialties. Moreover, 17.5% see taxonomy as the basis of scientific disciplines, 17.5% link taxonomy to the concepts of systematics and botany, and the least mentioned is the usefulness of the taxonomy (12%).

**Table 1.** Categorization of the university teachers' views on plant taxonomy and taxonomists.

| Views of the University Teachers | Reasoning Type | Percentage |
|---|---|---|
| -Because human survival depends largely on the knowledge of useful plants and toxic ones<br>-By definition, it is the science of classifying plants for a better understanding and enhancement of the potentialities and roles of species | Reasons related to the usefulness of the taxonomy | 12.0% |
| -Because taxonomy is a discipline of systematics both falling under general botany and systematic botany in particular<br>-The taxonomist deals with the classification of taxa according to various criteria. Taxonomy is a science that is inseparable from systematics<br>The taxonomist will make it possible to link plant taxa and group them with relevance.<br>-The taxonomist uses science for classification | Reasons linking taxonomy to notions of systematics and botany | 17.5% |
| -You have to be a scientist to practice taxonomy<br>-Systematics is a science in its own right that requires special scientific knowledge specific to this discipline.<br>-The taxonomist must have the tools and knowledge of scientific research<br>-The taxonomist must have the scientific basis to understand the logic used in taxonomy, and they must know the tools used to identify plants (insects and others) and be able to explain them and even use them with their students | Reasons related to the need to use scientific tools and scientific knowledge in taxonomy | 29.5% |

**Table 1.** *Cont.*

| Views of the University Teachers | Reasoning Type | Percentage |
|---|---|---|
| -It is impossible to practice taxonomy or plant systematics without having knowledge of plant biology, cellular and molecular biology, genetics, and computer science<br>-Taxonomy uses scientific techniques and disciplines<br>-Modern taxonomy uses the scientific method and is based on all the achievements of the different specialties of biology (genetics, cytology, ecology....)<br>-The taxonomy uses the scientific method and is based on the results of observation and experimentation, in harmony with the hypotheses of genetics and evolution | Reasons related to the use of scientific knowledge from other scientific specialties | 23.5% |
| -Taxonomy is the basis of all sciences, "how can we study a plant if we are not sure of its taxonomic position?"<br>-Taxonomy is the basis of all research work that requires exact identification of biological material<br>-Taxonomy is a science at the base of all scientific disciplines; in addition, this science uses data from molecular biology, genetics, etc. | Reasons related to considering taxonomy as the basis for scientific disciplines | 17.5% |

### 5.2. The University Teachers' Views on the Tasks of Plant Taxonomists and the Features of Taxonomy

To find out the university teachers' views on the tasks of plant taxonomists, inductive and deductive content analysis was used [81,82]. Although all the university teachers considered plant taxonomy as a scientific discipline, one out of six university teachers (16.7%) limited the tasks of a taxonomist to the nomenclature of species. They justified their answers by the following points: "The role of the taxonomist is to give a set of morphological, anatomical, karyological, palynological, chemical characters ... to these species". The majority thought that besides naming species, plant taxonomists also had other tasks (83.3%). Their reasons were diverse (Table 2).

**Table 2.** Classification of the university teachers' responses to plant taxonomists' tasks other than naming plant species.

| The Views of the University Teachers | Nature of the Reasonings | Percentage |
|---|---|---|
| -The taxonomist must be able to classify plant taxa in a systematic order<br>-The taxonomist will ensure the nomenclature by highlighting the affinities of a group.<br>-Only the individual has real existence. Taxonomy has created several conventional categories to present a classification accepted by the scientific community. These categories relate to the specific (= classification unit), infraspecific (subspecies, variety, form) and supra-specific (genus, tribes, family, order, class, branch, etc.) levels. Each level is defined by specific characters<br>-Nomenclature requires knowledge of botanical characters<br>-Because the question that arises is how to group all plants into species, genera and families on a scientific basis and why? | Reasons related to systematics | 35.8% |
| -Taxonomy is interested in other fields, such as biogeography, for example.<br>-Identification and description of species and their ranges<br>-It is necessary to have a good knowledge of the flora of a country and biology and ecology of plants<br>-Taxonomy is concerned with the behavior of species, their physiology, and their biochemical functions | Reasons related to Ecology | 21.4% |
| -Modern systematics is based on all the characteristics of plants, and there are many taxonomic criteria. Of course, the morphological and structural criteria are the basis, but we must also rely more on the biochemical and molecular criteria; the ultimate objective is knowledge for enhancement and preservation.<br>-Because a taxonomist must be multidisciplinary to study species (morphologically, genetically, etc.) before classifying and naming them. | Reasons related to phylogenetics | 21.4% |
| -Before the nomenclature of a plant, it is necessary to define its taxon, which requires the study of its characteristics and, in particular, those which are discriminating<br>-The nomenclature is the role of nomenclaturists | Other reasons | 21.4% |

University teachers' responses were various; they were grouped into four categories. The most mentioned justifications were based on systematics (35.8%), while the other three categories were fair (21.4% each): ecological rationales, phylogenetic reasoning, and general reasoning; this result is in agreement with the work of other researchers [16,17,19,67].

Table 3 shows that for the majority of the university teachers (78.3%), taxonomy is perceived as a strong scientific discipline, even if it is descriptive. They also stated that it is a discipline that evolves and is not static (86.4%).

**Table 3.** The university teachers' views on the features of taxonomy.

|  | Yes | No |
|---|---|---|
| -In your opinion, taxonomy is perceived as a purely descriptive and weakly scientific discipline? | 21.7% | 78.3% |
| -In your opinion, is taxonomy a static discipline? | 13.6% | 86.4% |

*5.3. Shortage of Taxonomists*

To find out the university teachers' views on the shortage of plant taxonomists, inductive and deductive content analysis was used [81,82]. The majority of the university teachers (96%) stated that the community of taxonomists is in danger. They explained the causes of this crisis by the factors presented in Table 4.

**Table 4.** Factors leading to the crisis of the taxonomic community.

| Factors | Percentage |
|---|---|
| -The lack of taxonomists/systematists | 82.60% |
| -The non-renewal of taxonomists/systematics (training and recruitment) | 76.40% |
| -The quality of the training provided at the university<br>-The first difficulties of taxonomists come from the general lack of knowledge about the plant world…<br>-Lack of specialized training (variation and evolution, systematics, taxonomy) at the level of a Master's degree focused on biodiversity conservation | 56.12%<br>56.12%<br>56.12% |
| -Disinterest of students in this discipline perceived as old-fashioned | 39.10% |
| -Nature of the plant taxonomy itself, because it is a difficult discipline, it requires more time in research to have significant results | 38.96% |
| -The disinterestedness of the ministries concerned and the lack of outlets.<br>-The ignorance of the leaders of the importance of this discipline<br>-Ignorance of what taxonomy is and of its interests in the knowledge, enhancement and conservation of biodiversity | 12.48% |
| Currently, the major problem of our students, all disciplines combined, is the lack of mastery of the language of instruction (French) | 4.16% |

The categorization of the university teachers' responses shows that the majority of the university teachers stated that the lack of taxonomists/systematicians and the absence of training and recruitment to renew taxonomists/systematicians are the main factors causing the lack of renewal of taxonomists (average of 80%); also the quality of training provided at the university remains insufficient or absent (56.12%). Two out of five university teachers also indicated the disinterest of students to plant taxonomy because they consider it an old-fashioned specialty and a difficult learning discipline. The university teachers also mentioned the ignorance of the actors responsible for this discipline, its importance and the disinterestedness of the ministries concerned, and the lack of outlets.

*5.4. Teaching Methods Used by the University Teachers in Plant Taxonomy*

To find out the teaching methods used by university teachers in plant taxonomy, inductive and deductive content analysis was carried out [81,82]. To question 16 in the questionnaire ("What teaching methods do you use for plant taxonomy courses?"), the university teachers who answered this question (16/24) gave different responses (Table 5).

**Table 5.** The university teachers' teaching methods in plant taxonomy.

| | |
|---|---|
| 1 | -Practical work with known species, then use of keys |
| 2 | -Give the morphological characters of each systematic group-course handout–lecture with a PowerPoint presentation—if the number of students is not large, we go out into the field |
| 3 | -Slideshows, handouts |
| 4 | -Base: handout-presentation of the main characters of the families studied |
| 5 | -Blackboard and chalk handout and documentaries |
| 6 | -In the form of theoretical and practical contents |
| 7 | -A lecture of about 20 h on the morphology of the majority of the main families of vascular plants in the spontaneous Moroccan flora; 6 h of tutorials to present the organography of vascular plants in the form of a descriptive sheet; 16 h of practical work to study the morphology of the main groups of vascular plants |
| 8 | -Lectures; Practical work and field trips for the preparation of herbariums for each theme |
| 9 | -Classical teaching methods |
| 10 | -Practical work and outings in the field |
| 11 | -Starting from examples and samples: morphological similarities first, the definition of the species and genus "Binomial taxonomy," then the groupings to go to the simplified classification table (Emb., Class, Family, genus and species), then we study the complex aspects of classification later |
| 12 | -Basic notions and working techniques, concrete examples with methodological approaches |
| 13 | -Projection of a summary on PPT, for example, showing the main characters differentiating the main groups in addition to descriptive handouts |
| 14 | -Observation and description of the different plant organs |
| 15 | -Interactive teaching method |
| 16 | -PowerPoint, student involvement in presentations, practical work in the classroom |

Most often were mentioned the traditional teaching methods (77.77%) (Table 6). Only 66.66% of the learning situations were based on practical work.

**Table 6.** Proportions of teaching methods used by the university teachers in plant taxonomy.

| Teaching Methods Used | Percentage |
|---|---|
| -Traditional teaching methods/lectures | 77.77% |
| -Laboratory works | 66.66% |
| -Tutorials | 33.33% |
| -Field trips and visits to greenhouses/gardens | 33.33% |
| -Interactive methods | 33.33% |
| -Use of documentaries | 16.66% |

Based on the answers to question 17 of the questionnaire ("Given a large number of students, what is the favorable method for teaching plant taxonomy?"), the university teachers consider the teaching methods presented in Table 7 as the most popular used teaching methods for a large part of students.

**Table 7.** Ranking, according to the university teachers, of the most popular teaching methods in the plant.

| Categorization of the Most Popular Teaching Methods | Percentage |
|---|---|
| Practical work in the classroom with a reduced number of students | 72.22% |
| Outing and visit of the gardens/Preparation of real herbariums | 38.88% |
| Use of ICT (TBI)/digital educational support/Preparation of virtual herbarium (photo)/Use of courses in digital platforms (Moodle) | 38.88% |
| Lectures in the amphitheater | 22.20% |
| Classroom tutorials | 11.10% |
| The great personal investment of students and interactive methods | 11.10% |
| Course handouts | 5.55% |
| Prerequisites and pre-university reinforce hourly volumes and expand the range of learning tools | 5.55% |

The majority of the university teachers recommended practical methods supervised by teachers to facilitate learning (72.22%). This supervision is necessary to easily resolve technical difficulties that may arise during learning situations. In order to target the quality of teaching instead of the quantity, which leads to the burden of the learning process and the demotivation of the students. The university teachers described teaching, e.g., in this way:

-the university teacher n° 9: "It will take at least 12 sessions of practical work to review the most important plant families in Morocco. The course should follow the content of the practical work and will content itself with giving a summary of the distinctive characteristics of each family. The course should be planned to become familiar with new phylogenetic methods".

-the university teacher n°11: "A large number of students require a large number of teachers, and the time necessary for learning plant taxonomy (currently 16 h of practical work) is derisory compared to the 56 h devoted to this discipline before the reform of the educational system which took place in 2008".

Almost two out of five university teachers affirmed that the effective teaching method is the methods which are based on outings and visits to the gardens and by preparing real herbariums; however, this can be scarcely achieved given the scarcity of taxonomy specialists on the one hand and plant material. Also, two out of five teachers propose the use of ICT (TBI) and the provision of digital teaching tools preparing virtual herbariums (photos) and using Moodle courses.

*5.5. Nature of Learning Difficulties in Plant Taxonomy and Enumeration of the Difficulties of Each Taxonomic Group*

The answers to question 18 make it possible to deduce the difficulties represented in Figure 1.

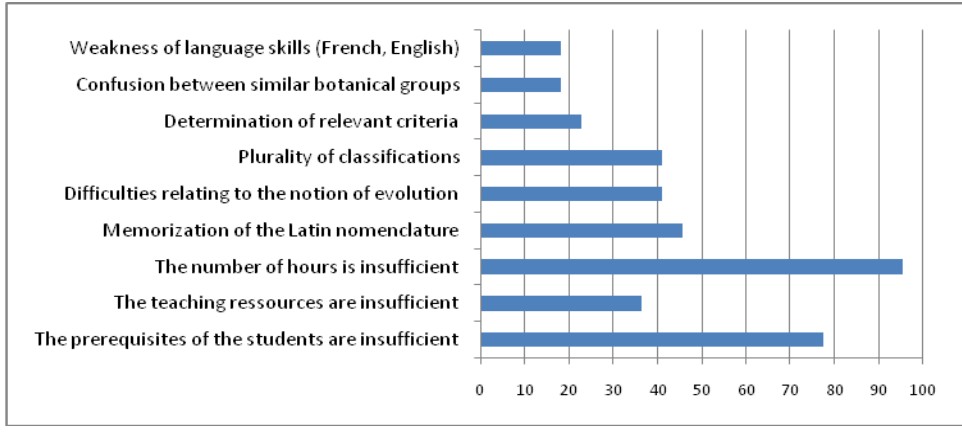

**Figure 1.** According to the university teachers, the types of difficulties that students encounter in learning plant taxonomy.

Almost all of the university teachers believed that the number of hours is insufficient and therefore constitutes one of the learning constraints. More than three out of four university teachers surveyed declare that the prerequisites of the students are weak and insufficient to continue this training at the university. An average of two out of five university teachers indicated that the difficulties are of a didactic nature, such as the means used in teaching and epistemological constraints relating to Latin nomenclature, typology of classifications, phylogenetics, and evolution. According to one-fifth of the university teachers, the determination of the characters of plant taxa and the confusion between similar taxonomic groups hinder learning among students. Three university teachers specified the problem of the constraint of hourly volume, especially for practical work, as perfectly insufficient. The university teachers also cited the lack of a botanical garden or arboretum-type structures, allowing the student to become familiar with the plant world. As for the prerequisites, the university teachers explained that it is necessary to have a pre-university base to facilitate plant taxonomy teaching. One participant wrote: "the lack of knowledge means that everything is new for the students and that they must learn by heart sometimes without previous knowledge related to botany and taxonomy". A university teacher also linked the learning difficulties to a "Lack of specialized teacher-researchers and lack of interest in the discipline in educational programs".

### 5.6. The Difficulties That Students Encounter concerning Taxonomic Groups According to University Teachers

Figure 2 represents the summary of the analysis of the university teachers' responses to question 19 in the questionnaire ("Which taxonomic group(s) is(are) the most difficult for the students?"). The university teachers confirmed that students encounter learning difficulties concerning different plant groups. 68.4% of them declared that students find difficulties in the angiosperm groups. A mean of one out of three teachers stated that students find difficulties in algae, bryophytes, and pteridophytes, while only 15.8% of teachers have noticed learning difficulties in gymnosperms.

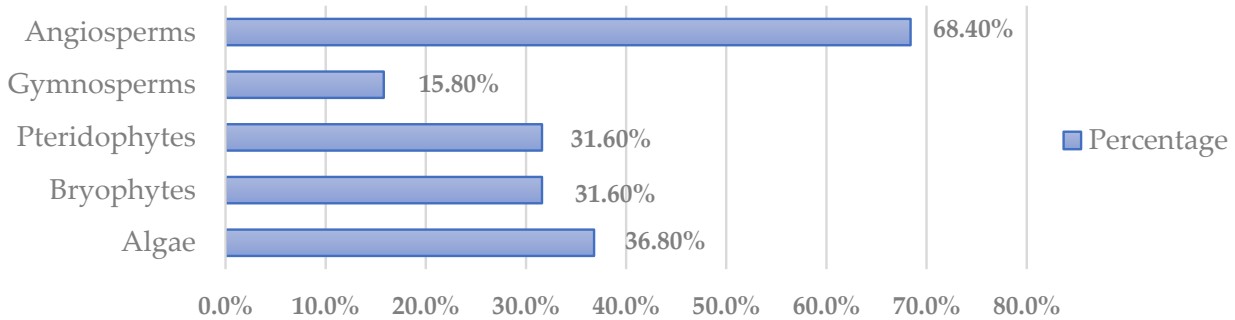

**Figure 2.** Classification of taxonomic groups according to their degrees of difficulty for students.

The responses to the question "why" make it possible to group university teachers' reasonings (Table 8).

There are four difficulty categories of plant groups:

-difficulty related to the diversity of groups, especially angiosperms

-difficulties related to the lack of knowledge

-technical difficulties related to the lack of plant material

-didactic difficulties related either to the knowledge taught or the Latin nomenclature to teachers' training quality.

The analysis of the questions on the difficulties encountered by the students in the taxonomic groups (Q20-in algae, Q21-in bryophytes, Q22-in pteridophytes, Q23-in gymnosperms and Q24-in angiosperms) (Table 9).

**Table 8.** Explanation of the difficulties of learning plant groups.

| Reasonings | Category | Percentage |
|---|---|---|
| -Angiosperms are the richest and most varied throughout the world and in Morocco.<br>-For the angiosperms, it is the multitude of identification criteria and the resemblance between certain groups that pose learning difficulties among students.<br>-The approach to Grasses and related groups (Juncaceae, Cyperaceae) by deviating from the usual model in terms of floral morphology remains the most difficult | Diversity | 40% |
| -Lack of knowledge and approach to life according to an evolutionary vision and a utilitarian function and problem of gaps in previous knowledge<br>-Because that are many gaps in students' previous botanical knowledge | Lack of knowledge | 25% |
| -Difficulty in collecting plants samples and insufficient materials for practical work, and lack of the necessary equipment<br>-Lack of data on Morocco for illustration in the case of Bryophytes. | Technical | 20% |
| -Teachers' training and quality of teaching/Latin nomenclature | Didactical | 15% |

**Table 9.** Difficulties encountered by students in learning about taxonomic groups.

| Taxonomic Groups | | | | |
|---|---|---|---|---|
| **Algua** | **Bryophyta** | **Pteridophyta** | **Gymnosperms** | **Angiosperms** |
| -The unavailability of some groups of fresh samples | -Scarcity and unavailability of some groups of fresh samples | -Scarcity and unavailability of some groups of fresh samples | -Rarity in Morocco and the non-availability of some groups of fresh samples | -The great diversity of this group and a great relatedness |
| -Lab material is insufficient<br>-lack of well-equipped lab rooms, delicate handling, and permanent need for the bench | -Lack of practical work and necessary equipment: practical workroom | -Lack of practical work and necessary equipment: practical workroom; A binocular magnifying glass is generally necessary for a good determination | -Lack of practical work and necessary equipment; A binocular magnifying glass is generally necessary for a good determination | -Lack of practical work and necessary equipment |
| -Difficulty of sample collection | -Difficulty remembering names/nomenclature Family/, Genus/; species/ | -Learn names and recognize species and remember the names of classes, families, genera and species | -Learn names and recognize species and remember the names of classes, families, genera and species | -Learn names and recognize species and remember names of classes, families, genera and species |
| -Determination and remembering the names of classes, families, genera and species<br>-The hourly load | -Disinterest and Lack of Observations | -Lack of interest | -Disinterest and lack of labeling in parks, for example, and incomplete keys, which leads to SP (unidentified species) | -Lack of interest and labeling in parks, for example, and lack of complete keys, which leads to SP (species not identified)<br>-Determination/description (Discover the distinctive characteristics of the different families) |
| -Lack of prerequisites | -Lack of knowledge | -Lack of knowledge | -Lack of knowledge | -Lack of knowledge<br>-Insufficient prerequisites and in Arabic |
| | -Reproductive cycle: the gametophyte is chlorophyllous | -Reproduction cycle and notions of spore versus seed | | Development cycle, fruits |

## 6. Discussion

According to Cintamulya and Mawartiningsih [27], information on plant species and skills to identify and classify plants are considered to be crucial issues for making inventories of living beings on the planet. Taxonomic information is essential for ecologists to monitor ecosystems and predict changes in biodiversity [14,16,17,53]. However, the number of taxonomy experts appears to be in decline [9,18,29]. In 1992, participants in the Conference on Biological Diversity highlighted three "taxonomic obstacles": the combination of large gaps in taxonomic knowledge, limited taxonomic infrastructure, and the decline of species experts [11,88]. The policy addressed the issue of the threat to biodiversity and the lack of information on living resources. The taxonomic hurdle interferes with the policy goal of making well-formed decisions regarding the conservation and use of biological resources [9,10]. However, the existence of impediments to taxonomy is recognized as one of the major obstacles to the conservation and sustainable use of biodiversity [88].

In this study, the focus is on the perceptions of university teachers on plant taxonomy and their opinions on the difficulties they face when teaching this subject and which have led to student disaffection and its link with the challenges of increasing the number of taxonomists.

The results showed that the university teachers surveyed considered a taxonomist as a scientist; nevertheless, their reasoning was different. The university teachers justified their view by saying that taxonomists use scientific tools and scientific knowledge in their work and also produce scientific information for the use of other disciplines, e.g., by combining taxonomy with plant systematics and other disciplines of botany [4,7]. They considered plant taxonomy as a dynamic, highly scientific, and descriptive discipline (cf. [2,23]). Their views were based mainly on systematics, ecology or phylogenetics. Thus, the justifications were different and depended on their educational paths and the nature of their university teaching, which reflects the diversity of the university teachers' views. Several previous studies have shown that individual conceptions can have effects on learning (e.g., [89,90]). For some, teachers' views can be a lever for the formation of new perceptions [91], while for others, they can minimize formative impact [92] or be an obstacle to professional development [93].

In addition to knowledge, the conception of the teacher is also a factor that influences their practice of learning in the classroom [94]. University teachers are no exception, and also their conceptions about teaching and learning influence their professional practices [95]. Research relating to the conceptions of higher education teachers about teaching and learning shows that, depending on how teachers envision teaching or learning, teachers' pedagogical choices are different [37,41,42]. Demougeot and Perret [37] assume that the considerations before the organization of their courses will not be the same. They add that "[f]or the less experienced, one can then think that the concerns about the preparation of their interventions in front of the students are linked to such conceptions" (p.9).

According to Ahrends et al. [11], botanists with little training were the least efficient in recording species, and botanists with an intermediate level of training were the most efficient [38]. In this study, university teachers stated that there are no official guidelines or official guides that stipulate the details of university teaching of plant taxonomy. The training guides are reference documents that serve to orient and supervise teachers, unify their visions about the activities of teaching plant taxonomy, and harmonize the teaching methods and techniques to improve teaching practices.

At the level of the biology departments, it is essential to prepare guides and reference documents intended for teachers of plant taxonomy and, if necessary, to qualify their ways of teaching [30,76]. They need guides that will therefore cover the various aspects of the pedagogical organization and the establishment of the physical and material resources required for this training [88,96]. This allows for the offering of teaching strategies that promote interaction in which the student commits to becoming an active learner. A study by Maskour et al. [73] showed that in Morocco, despite the majority of secondary school

teachers having studied the classification of plants during university, some do not consider the taxonomist a scientist, and others consider them so; however, the reasoning is different. Some students mentioned that in taxonomy, scientific knowledge and criteria are not used, that it does not use knowledge from other scientific specialties, or that it does not use the scientific approach [7,73]. The students had many misconceptions related to scientificity and the importance of plant taxonomy and its teaching, which partly explains the origin of the learning obstacles in plant taxonomy [68,71,72]. Several studies have shown that students have different misconceptions about plant taxonomy and its purposes, including distinguishing and clarifying the complex relationship between definition and classification activities. These misconceptions are identical to those mentioned in various studies [62–67]. Other research has identified difficulties in learning plant taxonomy in pupils and students [9,22,35,74]. They also revealed that didactic research relating to teaching methods for this biological discipline to improve is quite rare.

This study is consistent with other research on the fact that teaching plant classification is difficult and requires active methods centered on the student and on practical activities; these results are consistent with those of other researchers [9,22].

In this study, more than three out of four university teachers declared that the traditional methods dominate their teaching; these kinds of methods are unfavorable methods for the learning of the students [75,97]. The results also showed that one out of three university teachers does not rely on practical work. However, among their resolutions, the majority of them recommend practical methods supervised by supervisors to facilitate learning [49,58]. This supervision is necessary to easily resolve technical difficulties that may arise during learning situations and to target the quality of teaching instead of the quantity, which leads to the burden of the learning process and the demotivation of the students.

According to Wandersee and Schussler [98], several studies suggest that the declining interest in botany is actually due to the way botany is taught and presented around the world [59,60]. This could confirm that the traditional expository approaches that predominate the teaching of plant taxonomy at universities are not valid for the development of cognitive, strategic, and methodological skills. Moreover, the overload of the program and the quantity of information provided could constitute constraints for the students. Almost all university teachers declared that the number of hours allocated for teaching taxonomy is insufficient, especially for practical work [99]. Good management of the number of hours is essential because the time devoted to teaching is strongly linked to the learning of students [24,30,47]. This occurs on a linear basis: the more the percentage of schooling time increases, the more the percentage of knowledge retained by students increases [100]. Concerning the prerequisites, more than three of four university teachers declared that the prerequisites of the students are weak and insufficient to continue this training at the university. The lack of knowledge means that everything is new for the student and that they must learn by rote at times without previous knowledge. The pre-university education cycles must guarantee a base of prerequisites to facilitate education at universities [66,69,73]. Moreover, Uno [101] identified that plant biology is poorly covered in high school biology courses. He asserts that taxonomic misunderstanding can be attributed in part to a lack of relevance of plants to students' lives and also to the lack of previous knowledge related to plants and their classification. Bibeau [102] considers that learning difficulties are generally associated with problems of demotivation: the student can only learn and develop his skills if they are motivated to learn and if they make an effort to understand and carry out the activities proposed to him. Students feel held back when studying plant classification. This feeling of incompetence will negatively impact the motivational dynamics of students and their academic success [103].

In this study, the university teachers stated that the taxonomist community is in crisis, and the number of taxonomy experts seems to be in decline [7,33,88] due to the lack of taxonomists, the non-renewal of taxonomists at the training level, and terms of recruitment, and the quality of training provided at the university [7,11,33,104]. This

crisis is also linked to the lack of specialized teacher-researchers and the low importance given to this discipline in university programs. Some university teachers linked teaching difficulties to the weakness of scientific, didactic, and methodological skills. An average of two out of five university teachers presented that difficulties in the didactic order are the tools used in teaching and epistemological constraints relating to Latin nomenclature, typology of classifications, phylogenetics, and evolution [105]. According to one out of five university teachers, the students had difficulties in determining the characters of the taxa. They confuse similar plant taxonomic groups. This means that the skills of know-how relating to plant taxonomy are mediocre. This can aggravate the desire to learn this subject among students.

The Angiosperm group is the most difficult for students [106]. Similar difficulties were also mentioned by more than half of university students, and these difficulties are considered to make plant systematics a challenging discipline for students [62]. Likewise, the nature of the discipline itself calls upon other disciplines of biology. The limited and poor traditional teaching in practicum also contributes to teaching that favors memorization at the expense of reflection and understanding and to the lack of motivation of students for this discipline [43,73,103]. Learning plant taxonomy requires theoretical and practical teaching using diversification of didactic resources [88,96]. Thus, some authors have proposed that the teaching of plant classification should foster a form of active learning, emphasizing student-centered pedagogies such as group work to increase student engagement [43,107]. In addition, hands-on teaching based on the field environment helps expose students to living plants in their habitat and understand all forms of diversity, which would make students more motivated to learn [50–53]. Faced with the task of classifying plants, students are led to mobilize the knowledge previously studied during their school course.

To ensure the reliability of this study, the results have been compared with previous studies on plant classification teaching conducted in Morocco, as well as studies that have investigated students' learning difficulties, perceptions and misconceptions about plant taxonomy [66,67,69,72]. The research procedures were documented at different steps of the research, and the implementation was described in a comprehensive way. The qualitative data were analyzed using inductive and deductive content analysis. In addition, it was used both quantitative and qualitative approaches for providing a better understanding of the university teachers' views [108]. The study is consistent with other research based on the fact that teaching plant classification is difficult and requires active methods centered on the student and on practical activities [9,22].

## 7. Limitations of the Study

The results presented in the study cannot be generalized. First, the sample size of the study is small due to the fact that the study does not cover all Moroccan universities. Second, the answers given by university teachers are self-reported, so their answers are highly subjective. The results could also have been influenced in that the only data collection method was a questionnaire. In the future, alongside or instead of it, it would be good to use in-depth interviews to ensure a more versatile understanding concerning the respondents' views. The availability of equipment and teaching resources in the universities are not the same, which may also affect the views of the respondents.

## 8. Conclusions

Since plant taxonomy is a basic discipline in biology, it is the real foundation of biodiversity science and sustainable development. However, students' interest in plant taxonomy has declined significantly, leading to a reduction in their knowledge about plants, plant identification and use. At the same time, important plant information in terms of biodiversity and sustainable development has been lost. Similarly, the teaching of plant taxonomy and its university research, as well as the number of its experts, have decreased. This study focuses on the difficulties and causes that contribute to the decrease in learning plant taxonomy and the shortage of taxonomists based on the views of university teachers as

key stakeholders in the teaching and learning of plant taxonomy in Morocco. Based on the results, the university teachers considered a taxonomist as a scientist and plant taxonomy as a dynamic, highly scientific, and descriptive discipline. However, they justified their answers in different ways depending on their educational paths and the nature of their university teaching.

The teachers surveyed confirmed fully that all the financial, human, institutional, pedagogical, and didactic resources mobilized for the teaching of plant taxonomy are insufficient, which hinders the use of teaching methods supporting learning. They state that the community of taxonomists is in crisis because of the lack of taxonomists, the non-renewal of taxonomists, and the quality of training provided at the university.

Among the difficulties encountered by students, the views of the university teachers show that the students find the teaching of plant taxonomy insufficient, difficult and boring, and even worse, especially in the angiosperm group. The biggest challenges were the prevalence of traditional teacher-centered methods, the inadequacy of time and didactic resources spent on teaching, and the Latin nomenclature. The difficulties associated with the concept of evolution and the diversity of classifications were also mentioned.

The insufficiency of resources mobilized for the teaching of plant taxonomy, the misconceptions among students and their prerequisite knowledge scarcity about it leads to students' lowered motivation and disaffection for this branch and, consequently, to the reduction in the number of taxonomy specialists in Moroccan universities. It is essential to prepare guides and reference documents, and materials for teaching plant taxonomy. Teachers should also be taught teaching methods that lead students to experience active learning both inside and outside the classroom.

Finally, the results of this study can be used in education in relation to teaching and learning plant taxonomy worldwide. This knowledge in relation to plant identification is important not only for taxonomists, but also for many educators. Furthermore, it is also useful for biologists and other people interested in environmental issues, as it explains the importance of biodiversity for sustainability and development.

**Author Contributions:** Conceptualization, L.M., B.A., A.A.; Methodology, L.M., B.A. and E.J.; Data curation, B.E.B., J.K. and R.B.; Formal analysis, L.M., B.E.B.; and J.K.; Funding acquisition, L.M., B.E.B., A.A., E.J., B.A., J.K. and R.B.; Supervision, L.M. and E.J.; Validation, B.A., E.J. and A.A.; Writing—original draft, L.M.; Writing—review & editing, L.M., B.E.B., B.A., J.K., E.J., A.A. and R.B. All authors have read and agreed to the published version of the manuscript.

**Funding:** This research received no external funding.

**Institutional Review Board Statement:** This study followed the ethical principles of the Declaration of Helsinki in terms of confidentiality, anonymity and use of information for research purposes only.

**Informed Consent Statement:** Informed consent was obtained from all subjects involved in the study.

**Data Availability Statement:** Not applicable.

**Acknowledgments:** We acknowledge any support given which is not covered by the authors contributions.

**Conflicts of Interest:** The authors declare no conflict of interest.

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
