# Peer review of "Views of Moroccan University Teachers on Plant Taxonomy and Its Teaching and Learning Challenges"

_education, doi:10.3390/educsci12110799_

Round 1

Reviewer 1 Report

- Review several bibliographic references, which should be completed by adding the doi (Digital Object Identifier) : 

e.g. ref 35 :   Lindemann-Matthies, P. Investigating Nature on the Way to School: Responses to an Educational Programme by Teachers and 697 their Pupils. Int. J. Sci. Educ. 2006, 28(8), 895– 918.

add : https://doi.org/10.1080/10670560500438396

- References 32 and 33 have been introduced in the APA standard, need to be aligned with the others :

32. Fennane M., Ibn Tattou M., & El Oualidi J. (Eds), (2014). Flore pratique du Maroc. Travaux de l’Institut Scientifique, Rabat, Série 691 Botanique n° 40. Volume 3. 692

33. Fennane, M. & Ibn Tattou, M. (2012). Statistiques et commentaires sur l’inventaire actuel de la flore vasculaire du Maroc. Bulletin 693 de l’Institut Scientifique. Section Sciences de la Vie, n° 34, 1-9.

Reviewer 2 Report

Summary:

This article aimed to determine the different ways in which taxonomy is being taught in Moroccan universities as well as how students are influenced by the different teaching methods and other limitations in the progression of the field of taxonomy. The findings and recommendations made by the authors will positively influence the field of taxonomy and the way it is taught, which will have other positive effects in biodiversity research and conservation. 

Article: As a follow up to this research, it would be great to get the opinions of the actual students as well; to understand if they agree with the opinions of lecturers and to learn from their recommendations.

Review and specific comments:

Table 1 and Table 2 in the results should only be briefly summarized. The tables could perhaps be added as supplementary material. The focus of the paper should only be on improving the teaching and learning of taxonomy. The current overemphasis of "if taxonomists are considered scientists" and "what are the tasks of taxonomists" takes away from the rest of the work presented. The comments made about this topic in the discussion is sufficient and this topic should not take up so much of the results section.

Reviewer 3 Report

The importance of taxonomy to biodiversity is started several times in the first two pages.  A definition of taxonomy is also repeated.  These sentences become redundant and hamper the flow of the article.  I have listed the sentences that redundant.  Sentences 35,36,56,57 and 58, 81,82 and 83. 

The paragraph at sentences 75-79 states that taxonomists need to understand molecular biology, but it does not develop this idea or give examples.  This needs to be revised.

The article is clear and straight forward when it describes the state of taxonomy education in Morocco. The authors are on firm ground when they describe their experiment and results.  This section is clear and concise. 

The discussion is also clear and identifies the challenges perceived by university professors in teaching plant taxonomy.  They refer to taxonomists as him in line 475.  Taxonomy is not gender specific and that needs to be changed.  There is a paragraph at line 487 that is only one sentence long.  Paragraphs are more than one sentence long. 
